# Exploring the Antioxidant Potential of Phenolic Compounds from Winery By-Products by Hydroethanolic Extraction

**DOI:** 10.3390/molecules28186660

**Published:** 2023-09-16

**Authors:** Rui Dias Costa, Raúl Domínguez-Perles, Ana Abraão, Véronique Gomes, Irene Gouvinhas, Ana Novo Barros

**Affiliations:** 1Centre for the Research and Technology of Agro-Environmental and Biological Sciences (CITAB), Institute for Innovation, Capacity Building and Sustainability of Agri-Food Production (Inov4Agro), University of Trás-os-Montes e Alto Douro, Quinta de Prados, 5000-801 Vila Real, Portugal; anasantosa1@hotmail.com (A.A.); veroniquegomes@gmail.com (V.G.);; 2Phytochemistry and Healthy Foods Lab (LabFAS), CEBAS-CSIC, Campus Universitario de Espinardo, Edif. 25, 30100 Murcia, Spain; rdperles@cebas.csic.es

**Keywords:** winery by-products, (poly)phenols, antioxidant capacity, hydroethanolic extraction, HPLC-DAD

## Abstract

The residues generated in the wine industry (pomace, stems, seeds, wine lees, and grapevine shoots) are a potential source of bioactive compounds that can be used in other industries despite being sometimes underestimated. Different extraction methods using various solvents and extraction conditions are currently being investigated. Due to its natural occurrence in wines, safe behavior, and low toxicity when compared to other organic solvents, ethanol is used as an extracting agent. The aim of this study was to identify the winery by-product from the *Região Demarcada do Douro* and its corresponding extraction solvents that yields the most favorable results in (poly)phenols content and antioxidant capacity. To achieve this, five different ratios of ethanol: water, namely 0:100, 25:75, 50:50, 75:25, and 100:0 (*v*/*v*), for extracting the phenolic compounds were employed. Afterwards, the determination of total phenolic content (TPC), *ortho*-diphenols content (ODC), and flavonoid content (FC) as well as the antioxidant capacity of the obtained extracts using three different methods was performed. Since the best results of the spectrophotometric assays were obtained mostly with hydroethanolic extracts of stems (50:50, *v*/*v*), identification by HPLC-DAD has carried out. It was possible to conclude that the Tinta Roriz variety displayed the highest number of identified (poly)phenols.

## 1. Introduction

World wine production in 2023, excluding juices and musts, is projected to reach 258 million hectoliters (mhl), indicating a slight decline of nearly 3 mhl (-1%) when compared to the output in 2021 [1]. This production is accompanied by the generation of huge amounts of residues derived from wine processing. These include wastewater sludge, grape pomace (skins, pulp, and seeds), wine lees, grapevine shoots, and rachis, among others [2,3,4,5,6].

Considering the weight of grapes entering the winery, 83% correspond to the pulp, and the remaining 17% correspond to skins, seeds, stems, and lees, which are discarded [7]. Most of these discarded materials are used for composting, animal feed, and as fermentation substrates for biomass production [8,9,10]. However, if these wastes are not reused or discarded in the field without further treatment, they can cause negative environmental impacts [11,12]. Nevertheless, according to the emerging circular economy strategy, some of these by-products can be recycled, reused, or recovered, improving the sustainability and competitiveness of winemaking’s socioeconomic activity [13].

Some studies have already proven that winery by-products can be a source of natural antioxidants, mainly represented by polyphenols [2], that are able to promote human health according to the evidence gathered from both in vivo and in vitro research. Thereby, anticarcinogenic, anti-allergenic, anti-atherogenic, anti-inflammatory, anti-microbial, anti-degenerative, antioxidant, anti-thrombotic, cardioprotective, and vasodilatory actions are only a few of the biological properties that are attributed to phenolic compounds [10,14,15].

The phenolic compounds reported to date in these by-products include anthocyanins, proanthocyanidins and catechin derivatives, flavonols, phenolic acids, and stilbenes [16,17]. The quantitative profile of phenolic compounds in winery by-products depends on the type of grape (red or white), the tissue (leaves, stems, seeds, or peels), the processing conditions, grapevine variety, vine age, agroclimatic conditions of the production area, physicochemical conditions of the industrial process, the type of solvent, and the extraction procedures employed [10,18]. According to Nieto et al. [19], the extraction process constitutes a critical step for recovering bioactive compounds from plant materials. The chemical structure of the phenolic compounds extracted, the number and position of their hydroxyl groups, the molecular size as well as temperature, type of solvent, solvent composition, contact time, particle size, and interaction with other food ingredients can affect the extraction yield [20,21,22]. In this regard, it has been noticed that polar solvents are more effective at extracting polar chemicals, including most antioxidant compounds and phenolic compounds in winery by-products [2]. Although traditionally, water and organic solvents including acetone, ethanol, methanol, and ethyl acetate have been utilized in several conventional solid–liquid extraction procedures to extract phenolic compounds [22], hydroethanolic mixtures have been also broadly used for the extraction of phenolic compounds from winery by-products due to their high extractive yield [23,24,25,26,27,28,29,30,31,32,33]. In addition, the use of ethanol is more widespread than other solvents because it is generally recognized as safe (GRAS) by the Food and Drug Administration (FDA), is naturally present in the wine-making process, is allowed in the food industry, has a low boiling point (decreases energy costs), is easily eliminated from the extract by evaporation, is less consumed than other solvents, and presents low toxicity [21,24,34,35,36,37].

In this sense, the aim of this work was to analyze the extent in which eco-friendly solvents combinations allow enhancing the extraction of polyphenols from winery byproducts and select the best of them as a source of bioactive phenolics. The winery by-products, including stems, pomace, seeds, wine lees, and grapevine shoots, were collected from the three sub-regions of the *Região Demarcada do Douro* (Douro Superior, Cima Corgo, and Baixo Corgo). For this purpose, a solid–liquid extraction (SLE) using different water: ethanol combinations (0:100, 25:75, 50:50, 75:25, and 100:0, *v*/*v*) was applied to extract phenolic compounds from the winery by-products gathered. Afterwards, total phenolic content (TPC), *ortho*-diphenols content (ODC), and flavonoids content (FC) were set up resorting to spectrophotometric assays, and tentative identification by high-performance liquid chromatography (HPLC) coupled to photo diode array was performed. In addition, the antioxidant capacity was determined using two radical scavenging methods: 2,2-azino-bis (3-ethylbenzothiazoline-6-sulphonic acid) diammonium salt (ABTS) and 2,2-diphenyl-1-picrylhydrazyl (DPPH) and by the ferric-reducing/antioxidant power (FRAP) assay.

## 2. Results and Discussion

### 2.1. Phenolic Content of Winery By-Products

In this study, the extraction efficiency of five solvents combinations on the recovery of the phenolic compounds and antioxidant capacity from different WBP produced in the three sub-regions (Baixo Corgo, Cima Corgo, and Douro Superior) of *Região Demarcada do Douro* in northern Portugal was determined.

Samples are constituted by single cv. *Vitis vinifera* L. varieties or a mixture of them from the 2021 and 2022 harvests, of which the number of varieties present in each winery by-product is variable (ranging from two to thirteen).

The results obtained in this study regarding the phenolic content of all samples are presented in Figure 1, Figure 2 and Figure 3.

The three solvents combinations (SC) that presented the best results of total phenolic content (TPC) were SC2: ethanol: water (25:75, *v*/*v*), SC3: ethanol: water (50:50, *v*/*v*), and SC4: ethanol: water (75:25, *v*/*v*) (Table 1), with significant differences only between SC2 and SC3 for grape pomace (*p* < 0.05), grapevine shoots (*p* < 0.01), and grape seeds (*p* < 0.01). The use of SC3 showed the highest mean values of TPC in all WBP except for wine lees (Figure 1). On the other hand, the extractions with SC1: 100% H_2_O showed the lowest mean values except for stem samples. It is possible to conclude that for all WBP, the TPC values of the samples extracted with ethanol: water (50:50, *v*/*v*) are significantly different from those extracted with 100% H_2_O. Stem samples extracted with SC3 were the WBP with the highest mean values of TPC (61.12 ± 3.48 milligrams of gallic acid per gram of dry weight (mg GA/g DW), on average), significantly different (*p* < 0.01) from the extractions with SC1 and SC5. On the other hand, the wine lees extracted with the same solvent (SC3) were the samples with the lowest values (from 2.10 to 33.04 mg GA/g DW). Overall, the pomace extracted with SC1 was the sample with the lowest mean values (3.72 ± 2.59 mg GA/g DW).

In the case of *ortho*-diphenols content (ODC), SC3 represented the best results for stems (54.06 ± 3.28 mg GA/g DW, on average), grapevine shoots (17.81 ± 3.87 mg GA/g DW), and wine lees (from 5.11 to 39.65 mg GA/g DW) and the SC4 the best results for grape pomace (30.37 ± 3.08 mg GA/g DW) and seeds (from 9.69 to 61.69 mg GA/g DW). SC1 and SC5 expressed the lowest mean values of ODC. Furthermore, all seed samples did not show significant differences between solvent combinations The pomace extracted with SC1 was the WBP with the lowest mean values of ODC (4.46 ± 2.05 mg GA/g DW), generally (Figure 2).

Concerning the flavonoid content (FC), SC3 demonstrated the highest mean values for all WBP except for grape pomace, which showed the highest results with SC4, which was not significantly different from SC3. SC1 showed the lowest mean values for all WBP except for grape stems (Figure 3). It is also possible to observe that the highest values of FC were obtained for grape seeds extracted with SC3, the results of which ranged from 8.87 to 154.09 milligrams of catechin per gram of dry weight (mg CAT/g DW), and the lowest values were found in wine lees (from 0.82 to 20.79 mg CAT/g DW).

A work performed by Jimenez-Moreno et al. [25] that also used stems revealed that the better results were obtained with ethanol:H_2_O (50:50 *v*/*v*). In the study of Domínguez-Perles et al. [23], the authors used one similar proportion of SC3 in the same WBP, which was the most efficient condition for the extraction of the phenolic content. These values are in line with those of the present study. On the contrary, in the study of Sette et al. [38] that used SC1, the mean values of TPC in grape stems were lower (17.80 mg GA/g DW) than our results with the same solvent combination (40.30 ± 7.09 mg GA/g DW). In the same study, the TPC of stems was higher than that of pomace, as in our study.

Domínguez-Perles et al. [23] found that the phenolics compounds content could be decreased by higher or lower ethanol concentrations. In fact, we obtained the lowest values with 100% H_2_O (40.30 ± 7.09 mg GA/g DW) and 100% ethanol (16.82 ± 5.56 mg GA/g DW). According to Spigno and Faveri [24], a combination of solvents (such as ethanol) with water is more efficient for the extraction of phenolics from stems than with just a solvent.

Regarding grape pomace, in the study of Melo et al. [2], the results showed that the moderate concentration of ethanol (43% and 57%) was the best condition for the extraction of phenolic compounds. This proportion was similar to that of SC3. Pinton et al. [26] also obtained the best TPC using ethanol: water (40:60 *v*/*v*) and (60:40 *v*/*v*), with this last one being the proportion that produced higher TPC (27.48 mg GA/g DW) than our results using SC3 (from 7.01 to 33.01 mg GA/g DW). Once again, the use of mono-solvents (SC1 and SC5) produced the lowest TPC, as in our study for the pomace. The same authors obtained a lower mean value (8.31 mg GA/g DW) using SC1, and this value was higher than our result (Figure 1). According to the study of Sette et al. [38], the mean TPC in pomace extracted with water (2.50 mg GA/g DW) was lower than our results using SC1 (Figure 1). Makris et al. [27] used a similar proportion of SC3 (57% ethanol) to extract phenolic compounds of the pomace and obtained a high TPC (7.26 mg GA/g DW) and FC (7.22 mg CAT/g DW), which were lower than our results (Figure 1 and Figure 3).

Comparing our results with other extraction methodologies with the use of hydroethanolic solutions in pomace, in the study performed by Garrido et al. [28], 48% ethanol was the best optimal parameter found. In the study of Nayak et al. [21], the authors used a conventional extraction using water at 25 °C and obtained 4.28 mg GA/g DW of TPC and 3.01 mg GA/g DW of FC. These results were higher than ours using SC1 (Figure 1 and Figure 3). The results of TPC presented in the study of Tournour et al. [29] in pomace demonstrated higher values than those of the present study using ethanol: water (80:20, *v*/*v*) and SC1 (104.10 mg GA/g DW and 102.50 mg GA/g DW, respectively). The study of Pedras et al. [39], which used the subcritical water method with EtOH: H_2_O (25:75, *v*/*v*), obtained 47.30 mg GA/g DW of TPC, which was higher than the TPC of the pomace found in the present study (4.36 to 19.72 mg GA/g DW). According to the study of Makris et al. [18], the results of TPC in white grape pomace (48.26 mg GA/g DW) and in red grape pomace (54.02 mg GA/g DW) were higher than our results using SC3 (Figure 1).

Concerning seeds, the white grape seeds of the study performed by Makris et al. [18] revealed higher values of TPC in white and red grape seeds (111.08 mg GA/g DW and 103.30 mg GA/g DW, respectively) compared to our results using SC3 (Figure 1). Regarding FC, only the white grape seeds of this study showed higher results (Vio, MF, FP: 144.60 mg CAT/g DW) using SC3 compared to the study mentioned above (110.90 mg CAT/g DW). However, the authors used another extraction solvent. In the study of Medouni-Adrar et al. [30], ethanol was chosen as the best extraction solvent for seeds. In the same work, the highest TPC was detected in ethanol: water (50:50, *v*/*v*) with 86.51 mg GA/g DW. This result was higher than our results using SC3 (Figure 1). On the other hand, the authors used a similar ratio as SC4 (ethanol: water 74.33% *v*/*v*), which showed higher TPC results (96.23 mg GA/g DW) compared to the present work (from 10.03 to 77.11 mg GA/g DW) (Figure 1). The authors Shi et al. [40] also concluded that the extract with the highest total phenolics was the one obtained with SC3. In the study performed by Casazza et al. [41] in grape seeds that extracted phenolic compounds with ethanol for 19 h, the values of TPC and FC found were higher than our results using SC5. In the case of ODC, the values obtained by the authors (21.33 mg GA/g DW) were lower than our results Figure 2) using the same extraction solvent. Another study performed by Bucić-Kojić et al. [42] using a solid–liquid extraction with SC3 revealed values of TPC ranging from 14.72 to 66.81 mg GA/g DW, while our study presented values between 10.59 and 86.17 mg GA/g DW. In the study of Yilmaz et al. [31], the use of SC1 and SC5 was inadequate as the solvent for extraction of phenolics compounds from seeds, presenting the lowest values of TPC. In fact, extracts from seeds extracted with 50%, 60%, or 70% ethanol in water represented the highest TPC. These results are in line with the use of SC3 in our study, in which the best results were found with this solvent combination (from 10.59 to 86.17 mg GA/g DW). The same authors also mentioned that the mono-solvent was not as efficient in phenolic compounds extraction of grape seeds as an aqueous solution containing at least 50% of water due to the occurrence of glycoside derivatives of several phenolic chemicals found naturally in plant material, which make them more soluble in water, which is in accordance with previous studies [23,43].

With respect to wine lees, in the study performed by Romero-Díez et al. [44], they analyzed the TPC of first fermentation wine lees, obtaining 28.12 mg GA/g DW, using ethanol: water (50:50, *v*/*v*). This value was higher than our results (Figure 1). In another study of Romero-Díez et al. [45], the authors obtained the highest values of TPC (254 mg GA/g DW) and FC (146 mg CAT/g DW) using SC4. These results were higher than our results (Figure 1 and Figure 3). In the case of TPC, our study’s better results were achieved as well with SC4. Once more, the results of the work performed by Ciliberti et al. [46] were in line with our results since these authors obtained the highest TPC (38.56 mg GA/g) with SC3, and these values were higher than our results (from 2.10 to 33.04 mg GA/g DW). According to the literature, this is the first study to analyze the ODC in these kinds of matrices using these solvent combinations. In the case of stems and grapevine shoots, there are no studies in the literature that approach the FC.

Regarding the phenolic extraction of grapevine shoots, in the work performed by Çetin et al. [32] that used a solvent combination of EtOH:H_2_O (80:20 *v*/*v*) to extract phenolic compounds from Nebbiolo grapevine shoots, the values of TPC ranged from 25.36 to 36.56 mg GA/g DW, showing higher values than ours using SC4 (13.94–24.67 mg GA/g DW). On the contrary, in the studies of Moreira et al. [20], the results using SC3 were lower than those of the present work relative to TPC, with the exception of the Tinta Roriz variety (26.00 mg GA/g DW), whose results were higher than those of our Tinta Roriz sample (21.12 ± 0.52 mg GA/g DW). Once again, the best results of TPC in this by-product were obtained with SC3 in our study as well as in the study of Rajha et al. [33]. Shi et al. [47] found that the ethanol content of 50% (SC3) probably allowed the highest extraction yield due to the highest diversity of polyphenols. In addition, Goldstein and Chin [48] found that the polyphenols extraction using ethanol could influence cell permeability, which affects the phospholipid bilayer of biological membranes. The same authors concluded that the ethanol contents of 25% (SC2) and 75% (SC4) provided identical TPC. These results are in line with our results, with TPC values of 18.68 ± 3.09 and 19.76 ± 2.79 mg GA/g DW for SC2 and SC4, respectively. The results obtained by the same authors (10.00 mg GA/g DW) were lower than ours. Furthermore, the same authors concluded that SC1 showed the worst results, which is in line with the results obtained in the present work (Figure 1). The TPC values of the study performed by Alexandru et al. [49] (around 50 mg GA/g DW), who used maceration with 100% ethanol for 24 h at room temperature to extract phenolic compounds from Nebbiolo grapevine shoots, were higher than our values (15.06 ± 3.35 mg GA/g DW).

With regard to the phenolic extraction in other plant material, the study of Schechtel et al. [50] that extracted phenolic compounds in *Flaxleaf Fleabane* leaves revealed the best results of TPC, FC, and ODC using SC3 and SC4. The results of these authors are in line once again with our results. In most of the analyses, the lowest FC values were obtained with SC1 except in stems, which showed lower results with SC5 (Figure 3). Another study performed by Escher et al. [51] that analyzed the chemical composition of *Calendula officinalis* flower extracts showed that the hydroalcoholic extract (50:50, *v*/*v*), corresponding to SC3, presented the highest total phenols and flavonoids content. Despite the fact that these results are related to different types of samples, they are consistent with the findings of the current investigation.

### 2.2. Antioxidant Capacity of Winery By-Products

The antioxidant capacity of WBP samples was determined by the ferric-reducing antioxidant power (FRAP) and by two radical scavenging methods, namely DPPH and ABTS.

Regarding the results of antioxidant capacity by FRAP (Figure 4), the best results were obtained with SC3 in the case of stems, pomace, wine lees, and grapevine shoots. On the other hand, using SC4, we obtained better results in seeds, with no significant differences between both solvents combinations. The pomace, seeds, and grapevine shoots revealed lower values with SC1, and stems and wine lees presented lower values for these methodologies when extracted with SC5. As in FC, only in grape pomace were there significant differences between SC1 and SC5 (*p* < 0.05). In the cases where SC3 showed better results and SC5 lower values (grape stems and wine lees), there were significant differences between the combinations. Upon measuring the antioxidant capacity using FRAP methodology, the pomace extracted with SC1 was also significantly different from the one obtained by the use of SC3 (*p* < 0.001). The grapevine shoots and seeds that obtained better results with SC4 were significantly different from the combination that showed the lowest values (SC1): *p* < 0.01 and *p* < 0.001, respectively.

The antioxidant capacity of stems extracts obtained with extraction using ethanol: water (50:50, *v*/*v*) were the samples with the highest values (0.58 ± 0.03 millimoles of Trolox per gram of dry weight (mmol T/g DW), on average) of the FRAP assay. On the other hand, the grape-vine shoots showed the lowest values (0.20 ± 0.05 mmol T/g DW) (Figure 4).

Concerning the results of the antioxidant capacity referring to the inhibition of DPPH and ABTS radicals for the majority of the WBP studied (Figure 5 and Figure 6), it was possible to observe that extractions with SC3 and SC4 showed the best performances, with no significant differences between them. It is noteworthy that in the case of pomace and grapevine shoots extracted with SC1, they showed the lowest mean values of antioxidant capacity using both methodologies, and the differences between this solvent combination with SC3 and SC4 were significantly different. Once more, the stems extracts obtained with SC4 were the samples that revealed the best results of DPPH (0.55 ± 0.06 mmol T/g DW), and the stems extracts achieved with SC3 showed the best results of ABTS (0.56 ± 0.03 mmol T/g DW). On the contrary, the wine lees extracted with SC5 exhibited the lowest mean value of DPPH (from 0.00 to 0.10 mmol T/g) (Figure 5), and the pomace extracted with SC1 showed the lowest mean value of ABTS (from 0.02 to 0.05 mmol T/g DW) *(*Figure 6).

Furthermore, using SC3, the grape-vine shoots were the WBP with the lowest values for FRAP (0.20 ± 0.05 mmol T/g) (Figure 4). Regarding the DPPH methodology, the wines lees were the samples with the lowest value (Figure 5). In the case of ABTS assay, using the same solvents combination, the pomace presented the lowest value of ABTS (from 0.04 to 0.24 mmol T/g). It was possible to conclude that stems extracted with SC4 displayed the best antioxidant capacity performance in the FRAP assay and ABTS assay applied (Figure 4 and Figure 6).

Regarding some studies that determined the antioxidant capacity of stems extracts using the same solvents combinations, Schechtel et al. [50] demonstrated that the values of FRAP and DPPH were better when stems extracts were prepared with SC3 and SC4, as in our study. For most of the antioxidant capacity analysis, the lowest values were obtained with SC1, as in our study, except for grape stems (FRAP, DPPH, and ABTS), wine lees (FRAP and DPPH), and seeds (ABTS), which showed lower results with SC5. The results of the study performed by Jimenez-Moreno et al. [25] also revealed higher antioxidant capacities by FRAP when SC3 was used. These results are in line with those of the present study.

Regarding grape pomace, our results from measuring antioxidant capacity by DPPH methodology using all solvents combinations were lower than the results demonstrated by Tournour et al. [29], who used an ethanol/water (80:00 *v*/*v*) (0.810 mmol T/g DW). The same happened with the use of SC1, in which the authors obtained 0.860 mmol T/g DW. In spite of the results being expressed in different units, Nayak et al. [21] obtained higher DPPH values with the use of SC3 compared to SC1. Pinto et al. [26], who worked with conventional solvent extraction (60% ethanol), obtained ABTS values higher (3.01 mmol T/g DW) than those of the present study using SC3 (from 0.04 to 0.24 T/g DW) in the case of pomace (Figure 6).

In the case of seeds, the DPPH and ABTS results (0.04 and 0.07 mmol T/g DW, on average) obtained in the study of Ky et al. [52], which used water: ethanol (95:5 *v*/*v*) with chloroform to remove lipophilic material, were lower than our values using 100% ethanol, as can be seen in the Figure 5 and Figure 6.

Concerning wine lees, Romero-Díez et al. [45] showed higher results for FRAP assays using SC3 (2.11 mmol T/g DW) and SC4 (2.20 mmol T/g DW) compared to those of the present study (Figure 4), with no significant differences between them. In our case, comparing the results with SC3 and SC4, the results are lower (Figure 4) and not significantly different. In spite of the work performed by Ciliberti et al. [46], who used another extraction method, the authors concluded that the use of ethanol: water (50:50, *v*/*v*) revealed higher FRAP values compared to the use of water, which is in line with our values. In the case of ABTS, the authors obtained the best results with the use of water extracts. The ABTS results of our study using SC1 and SC3 (Figure 6) are lower than the values of these authors (1.23 and 0.42 mmol T/g DW, respectively).

Regarding the antioxidant capacity of grapevine shoots, in the study of Moreira et al. [20], the authors obtained, for radical scavenging capacity using DPPH method, lower results (Touriga Nacional: 0.03 mmol T/g DW, Tinta Roriz: 0.06 mmol T/g DW) compared to our samples (0.18 ± 0.05 mmol T/g, on average) extracted with SC3. Comparing the Tinta Roriz analyzed in both works with the same solvent combination, we obtained a higher mean value (0.21 ± 0.01 mmol T/g DW).

According to our knowledge, this is the first study to analyze the antioxidant capacity by FRAP methodology in pomace, seeds, and grapevine shoots using these solvent combinations. In the case of the antioxidant capacity determined by DPPH assay, there are no studies in wine lees also using these solvent combinations. On the other hand, no studies were found regarding the antioxidant capacity determined by ABTS methodology in stems and grapevine shoots.

### 2.3. Principal Component Analysis

PCA is a very useful technique that allows the compression of information from many variables into a few uncorrelated variables known as principal components (PCs). PCA has been widely used in a variety of areas and fields, including distinguishing bioactive constituents and targeting them to specific bioactivities. Given that stems were the by-product exhibiting the best performances regarding phenolic content and antioxidant capacity, a PCA analysis was performed in order to verify how the samples cluster according to the solvent combinations used.

Figure 7 showed the scatter plot of PCA applied to the phenolic content and antioxidant capacity assays in stem samples. The first two-dimensional components explained 96.38% and 2.04% of the loading score, respectively. In the upper and lower left quadrants, the group of the SC1 and SC5 represents the lowest values of all parameters. On the other hand, in the right side of the plot, it is possible to observe the stem samples that correspond to SC2, SC3, and SC4, characterized by having, in general, the highest values of all parameters studied. It is possible to conclude that they have opposite responses compared to SC1 and SC5.

### 2.4. Phenolic Content and Antioxidant Capacity of Stem Varieties Hydroethanolic Extracts (50:50, v/v)

Since the stem extracts extracted with SC3 yielded the most favorable results, we now delve into a comprehensive discussion of the findings for each stem variety.

As evident from Table 1, among the white varieties, Verdelho stands out with the highest values for TPC, FC, FRAP, and ABTS results. On the other hand, it is noteworthy that the Tinta Roriz variety emerged as the leading red grape variety, exhibiting the highest results in all spectrophotometric assays except for the ODC. Only in TPC and in FRAP was Tinta Roriz significantly different compared to Touriga Franca.

For these results, we chose this winery by-product to gain a deeper understanding of its (poly)phenolic composition.

### 2.5. Phenolic Compound Identification in Hydroethanolic Stem Extracts (50:50, v/v) Using HPLC-DAD

Given that stems were found to exhibit the highest values of TPC, ODC, and antioxidant activity when utilizing the ethanol: water solvent combination (50:50, *v*/*v*), we proceeded to the analysis of all stem samples using high-performance liquid chromatography with diode array detection (HPLC-DAD). Table 2 presents the detailed identification of phenolic compounds found in the stem extracts, while Figure 8 showcases a representative chromatogram obtained through HPLC-DAD.

The red variety Tinta Roriz stood out as the variety with the highest number of phenolic compounds identified, making a total of 27. In turn, Códega do Larinho was the variety where the fewest phenolic compounds were identified (Table 2).

#### 2.5.1. Non-Flavonoids

##### Phenolic Acids

In this study, two phenolic acids were identified: protocatechuic acid hexoside (peak 1) and *trans*-caftaric acid (peak 3). Both compounds were identified in all stem varieties with the exception of Verdelho, where protocatechuic acid hexoside was not detected.

Our identification is consistent with the findings of Anastasiadi et al. [53], who also observed the presence of *trans*-caftaric acid in their stem samples.

As of our current knowledge, no prior studies have reported the identification of protocatechuic acid hexoside in stems.

##### Stilbens

In the stems varieties investigated within study, three stilbenes were identified: oxyresveratrol (peak 16), *trans*-piceid (peak 19), and ԑ-viniferin (peak 26) (Table 2).

Our findings are in line with previous research conducted by Nieto et al. [19], Sun et al. [54], and Costa-Pérez et al. [17], who also identified *trans*-piceid in stems.

Regarding ε-viniferin, it was also identified in stems varieties analysis conducted by Gouvinhas et al. [4], Dias et al. [11], Leal et al. [12], Barros et al. [55], and Esparza et al. [56].

In addition to the previous compounds, oxyresveratrol was also identified in the study conducted by Costa-Pérez et al. [17].

#### 2.5.2. Flavonoids

##### Flavanols

In the current study, two flavanols were identified in the stem varieties: catechin (peak 6 and epicatechin gallate (peak 12).

Catechin was also found in the studies performed by Leal et al. [12], Costa-Pérez et al. [17], Esparza et al. [56], and Prusova et al. [57].

In the case of epicatechin gallate, its presence was also reported by Nieto et al. [19] and Anastasiadi et al. [53].

##### Flavonols

The present study successfully identified a total of five flavonols in the stem varieties. These flavonols comprise quercetin-glucoside (peak 9), quercetin-3-rutinoside (peak 11), quercetin-3-*O*-glucuronide (peak 13), kaempferol-3-*O*-glucoside (peak 14), and kaempferol-7-*O*-β-d-glucopyranoside (peak 18).

In the case of quercetin-glucoside, it was identified in work of Jiménez-Moreno et al. [25].

Regarding quercetin-3-rutinoside, Gouvinhas et al. [4] identified also this flavonol using RP-HPLC-DAD.

Quercetin-3-*O*-glucuronide was identified in the studies of Gouvinhas et al. [4], Barros et al. [55], and Dias et al. [11].

On the other hand, kaempferol-3-*O*-glucoside was also identified by Gouvinhas et al. [4], Barros et al. [55], Dias et al. [11], and Souquet et al. [58].

As far as our knowledge extends, the presence of the compound kaempferol-7-*O*-β-D-glucopyranoside has not been observed in stems.

##### Flavones

Among the flavone class of compounds, the luteolin-rutinoside (peak 15) was the sole representative identified in three distinct stem varieties (Table 2). To the best of our knowledge, this compound has not been reported in previous studies involving grape stems.

##### Anthocyanins

In this study, a total of eleven anthocyanins were successfully identified in the stem varieties: delphinidin-3-*O*-glucoside (peak 17), cyanidin-3-*O*-glucoside (peak 20), peonidin-3-*O*-glucoside (peak 23), malvidin-3-*O*-glucoside (peak 24), delphinidin-3-*O*-acetylglucoside (peak 25), peonidin-3-*O*-acetylglucoside (peak 27), malvidin-3-*O*-acetylglucoside (peak 28), delphinidin-3-*O*-*p*-coumaroylglucoside (peak 29), cyanidin-3-*O*-*p*-coumaroylglucoside (peak 30), petunidin-3-*O*-*p*-coumaroylglucoside (peak 31), and malvidin-3-*O*-*p*-coumaroylglucoside (peak 32) (Table 2).

The variety Folgasão did not exhibit any detected anthocyanins. However, in contrast, the Tinta Roriz variety showed the presence of all eleven identified anthocyanins (Table 2).

In comparison with other studies, Nieto et al. [19] reported the quantification of delphinidin-3-*O*-glucoside and cyanidin-3-*O*-glucoside. Additionally, Nieto et al. [19], Barros et al. [55], and Dias et al. [11] identified malvidin-3-*O*-glucoside.

According to the available literature, the remaining anthocyanins were not found in the stem samples investigated in previous studies.

#### 2.5.3. Proanthocyanidins

This study revealed the presence of seven proanthocyanidins, proanthocyanidin dimer (B-type) isomer 1 (peak 2), proanthocyanidin dimer (B-type) isomer 2 (peak 4), proanthocyanidin trimer (B-type) isomer 1 (peak 5), proanthocyanidin dimer-gallate isomer 1 (peak 7), proanthocyanidin dimer-gallate isomer 2 (peak 8), proanthocyanidin trimer (B-type) isomer 2 (peak 10), and proanthocyanidin trimer (B-type) monogallate (peak 21), as presented in Table 2. Both the Verdelho and Malvasia Fina varieties exhibited the identification of all these compounds.

The proanthocyanidins corresponding to the peaks 2, 4, 5, and 10 were similarly identified in the study conducted by Costa et al. [17].

Based on the available literature, the stem samples investigated in previous studies did not reveal the identification of the remaining proanthocyanidins by HPLC-DAD.

## 3. Materials and Methods

### 3.1. Chemicals and Reagents

The potassium hydroxide, Folin–Ciocalteu’s reagent, gallic acid (3,4,5-trihydroxybenzoic acid) and acetic acid (both extra pure (>99%)), and sodium hydroxide (98%) were purchased from Panreac (Panreac Química S.L.U., Barcelona, Spain). Sodium nitrite, aluminum chloride, and sodium carbonate (all extra pure (>99%)) and ethanol were purchased from Merck (Merck, Darmstadt, Germany). Sodium molybdate (99.5%) was purchased from Chem-Lab (Chem-Lab N.V., Zedelgem, Belgium).

Additionally, catechin (98%), Trolox (6-hydroxy-2,5,7,8-tetra-methylchroman-2-carboxylic acid, ≥98.0%), DPPH^•^ (2,2-diphenyl-1-picrylhidrazyl radical, ≤100.0%), ABTS^•+^ (2,2-azino-bis (3-ethylbenzothiazoline-6-sulphonic acid) diammonium salt ≥ 98.0%), potassium persulfate (K_2_S_2_O_8_, ≥99.0%), TPTZ (2,4,6-Tripyridyl-s-Triazine, ≥98.0%), and iron (III) chloride (FeCl_3_) (≥99.9%) were obtained from Sigma-Aldrich (Steinheim, Germany). Distilled water (Millipore, Bedford, MA, USA) was used for all extractions and analyses.

Formic acid was obtained from Panreac (Castellar del Vallés, Barcelona, Spain). Acetonitrile was provided by J.T. Baker (Philipsburg, NJ, USA).

### 3.2. Sampling

The WBP were collected in the 2021 and 2022 harvests at two different winery industries (Rozés and Adega de Vila Real) and one viticulture company (Daniel Fernandes—Unipessoal de Viticultura) from the sub-regions of the *Região Demarcada do Douro*. As shown in Table 3, a total of 28 samples of WBP were used, including six stem samples, three grape pomace samples, two wine lees samples, thirteen grapevine shoots samples, and four seeds samples.

### 3.3. Preparation of Winery By-Products Extracts

The schematic representation of the methodologies employed in the present study is illustrated in Figure 9. The extraction of phenolic compounds was carried out according to Abraão et al. [59] with some modifications. To perform the extracts, the stems, whole pomace, and seeds were dried in an oven (Memmert, Schwabach, Germany) for 72 h at 40 °C. In the case of the wine lees and grapevine shoots, they were frozen at −80 °C and then lyophilized (VirTis Benchtop Pro Freeze-drier with OmnitronicsTM, SP industries, Inc, Warminster, PA, USA). Three extracts of each sample were prepared by weighing 40 mg of the previously milled sample, and then, 1.5 mL of the five extracting solvents were added (solvent combination (SC1: (100% H_2_O), SC2: EtOH: H_2_O (25:75, *v*/*v*), SC3: EtOH: H_2_O (50:50, *v*/*v*), SC4: EtOH: H_2_O (75:25, *v*/*v*), and SC5: 100% EtOH). The mixtures were thoroughly homogenized and agitated for 30 min in an orbital shaker (GFL 3005, GEMINI, Apeldoorn, The Netherlands). Then, the mixtures were centrifuged (Sigma 2-16KL Refrigerated Centrifuges, Sigma Laborzentrifugen, Berlin, Germany) at 10,000× *g* for 15 min at 4 °C, and finally, the supernatants were collected. This procedure was repeated three times. The resulting extraction volumes (supernatants) were adjusted with the five extracting solvents to 5 mL in a volumetric flask (OlimPeak, Teknokroma, Barcelona, Spain) and then filtered using 0.2 m regenerated cellulose filters. In the case of the grape pomace and wine lees, the solvent combinations were acidified with 0.1% hydrochloric acid.

### 3.4. Determination of Phenolic Content

The phenolic content of WBP extracts were determined by spectrophotometric methodologies adapted for 96-well microplates (PrimeSurface MS-9096MZ, Frilabo, Maia, Portugal) according to Gouvinhas et al. [4], with some modifications. The absorbances were measured using microplate readers (Multiskan GO Microplate Photometer, Thermo Fisher Scientific, Vantaa, Finland).

#### 3.4.1. Total Phenols Content

The total phenols content (TPC) of WBP extracts was determined by adding 20 μL of the sample to 100 μL of Folin–Ciocalteu reagent. Then, 80 μL of Na_2_CO_3_ (7.5%) was added. The reaction was incubated in an oven at 40–45 °C for 30 min and protected from light. Absorbance was measured at 750 nm. Gallic acid was used as a standard, and the results are expressed in mg of gallic acid per gram of dry weight (mg GA/g DW) using gallic acid as standard.

#### 3.4.2. *Ortho*-Diphenols Content

The *ortho*-diphenols content of WBP extracts was determined by adding 40 μL of Na_2_MoO_4_ (50 g/L) to 160 μL of the samples appropriately diluted. Mixtures were vortexed and allowed to rest at room temperature, protected from light, for 15 min. The absorbance was measured at 375 nm and quantified using gallic acid as standard. Results are expressed in mg GA/g DW.

#### 3.4.3. Flavonoids Content

The flavonoid content (FC) of samples extracts was measured based on the formation of a flavonoid–aluminum complex. Firstly, 24 μL of the diluted sample was mixed with 28 μL of NaNO_2_ (50 g/L). After exactly 5 min, 28 μL AlCl_3_ (100 g/L) was added, and the mixture was allowed to react for 6 min. Finally, 120 μL of NaOH (1 M) was added to the mixture. The absorbance was immediately measured at 510 nm. Catechin was used for the construction of the calibration curve. Results are expressed in mg of catechin per gram of dry weight (mg CAT/g DW).

### 3.5. Determination of Antioxidant Capacity

The antioxidant capacity was determined using three different spectrophotometric methodologies, namely FRAP, DPPH, and ABTS^•+^, according to Santos et al. [60], with some modifications.

#### 3.5.1. Antioxidant capacity by Ferric-Reducing Antioxidant Power (FRAP)

For FRAP assay, 20 µL of the extracts were mixed with 180 µL of FRAP working solution (1 volume of TPTZ (10 mM dissolved in hydrochloric acid), 1 volume of ferric chloride (20 mM in water), and 10 volumes of acetate buffer (300 mM, pH 3.6)). Then, the microplate was incubated at 37 °C for 30 min and protected from light. After, the absorbance was read at 593 nm, and Trolox was used as a standard, with the results expressed in mmol Trolox/g DW.

#### 3.5.2. Antioxidant Capacity by DPPH

Regarding the DPPH radical scavenging assay, 10 µL of the extract and 190 µL of the DPPH working solution, previously prepared, were mixed and reacted for 30 min protected from light at room temperature. After 30 min, the absorbance at 520 nm was read with 70% hydroethanol used as blank. The scavenging capacity of the samples was calculated by the interpolation of the Trolox calibration curve, and the results are expressed in mmol Trolox/g DW.

#### 3.5.3. Antioxidant Capacity by ABTS

For the determination of antioxidant capacity by ABTS, initially, 12 µL of each winery by-product extract were blended with 188 µL of ABTS working solution (5 mL of ABTS stock solution (7.0 mM in water) with 88 µL of potassium persulfate (148 mM) and diluted to a working solution with sodium acetate buffer (20 mM, pH 4.5)) and left to react, protected from light. One well with 188 µL of ABTS solution work and 12 µL of distilled water was used as a blank. After 30 min, the absorbance was measured at 734 nm and quantified using Trolox as standard. Results are expressed in mmol Trolox/g DW.

### 3.6. Identification of Phenolic Compounds by HPLC-DAD

To gain a more comprehensive understanding of the extract’s composition, we employed HPLC-DAD (high-performance liquid chromatography with diode array detection) according to Costa-Pérez et al. [17]. The chromatographic separation of the phenolic compounds present in the analytical extracts was carried out using an Agilent HPLC 1100 series equipped with a diode array detector (Agilent Technologies, Waldbronn, Germany). The separation was performed on a Luna C18 column (250.0 × 4.6 mm, 5.0 μm particle size, Phenomenex, Macclesfield, UK). The HPLC system comprised a binary pump (model G1312A), an autosampler (model G1313A), a degasser (model G1322A), a photodiode array (PDA) detector (model G1315B), and an ion trap spectrometer (model G2445A). The LCMSD software (v. 4.1, Agilent Technologies) controlled the operation according to the chromatographic specifications described by Barros et al. [55]. Water/formic acid (99:1, *v*/*v*) was employed as solvent A, while acetonitrile/formic acid (99:1, *v*/*v*) was used as solvent B for chromatographic separation. Spectral data from all peaks were recorded in the 200–600 nm range, allowing for a comprehensive analysis of the phenolic compounds present in the analytical extracts. Chromatograms were recorded at 280 nm for proanthocyanidins, 320 nm for phenolic acids and stilbenes, 360 nm for flavonols, and 520 nm for anthocyanins.

### 3.7. Statistical Analysis

All the results are presented as mean ± standard deviation (SD) for the determination in triplicate. Statistical comparisons were made using the nonparametric Friedman’s test (IBM SPSS Statistics 27) to detect differences between the extraction solvent proportions of winery by-products in terms of TPC, ODC, FC, FRAP, DPPH, and ABTS assays. Significance values were adjusted by the Bonferroni correction for multiple tests. Also, it was performed an analysis of variance (ANOVA) followed by a post-hoc Tukey to detect differences between the hydroethanolic extracts (50:50, *v*/*v*) of stem varieties.

A principal component analysis (PCA) was performed using the mean values of the triplicates in the MATLAB R2019b environment (MathWorks, Inc., Natick, MA, USA). The data were adjusted to a range of 0–100 while accounting for the highest mean value determined during each experiment.

## 4. Conclusions

Considering the aim of this research, the hydroethanolic extracts of stems (50:50, *v*/*v*) demonstrated potential as a rich source of phenolic compounds, exhibiting the highest values of phenolic content and antioxidant capacity as measured by spectrophotometric assays. It is possible to conclude that the stems are a source of (poly)phenols, including phenolic acids, flavonols, flavanols, a flavone, anthocyanins, stilbenes, and proanthocyanins. Tinta Roriz was the variety with more phenolic compounds identified by HPLC-DAD and the highest values of TPC, FC, and antioxidant capacity. Based on the findings from this study, it is evident that this winery by-product can be regarded as a highly promising and rich source of natural bioactive compounds with antioxidant potential, namely the Tinta Roriz variety. As a result, it emerges as a strong candidate for future applications in the food, cosmetic, and/or pharmaceutical industries, contributing to the circular economy and industrial symbiosis. Further studies should be performed to quantify all (poly)phenolics identified in this by-product and correlate them with antioxidant capacity and other biological activities.

## Figures and Tables

**Figure 1 molecules-28-06660-f001:**
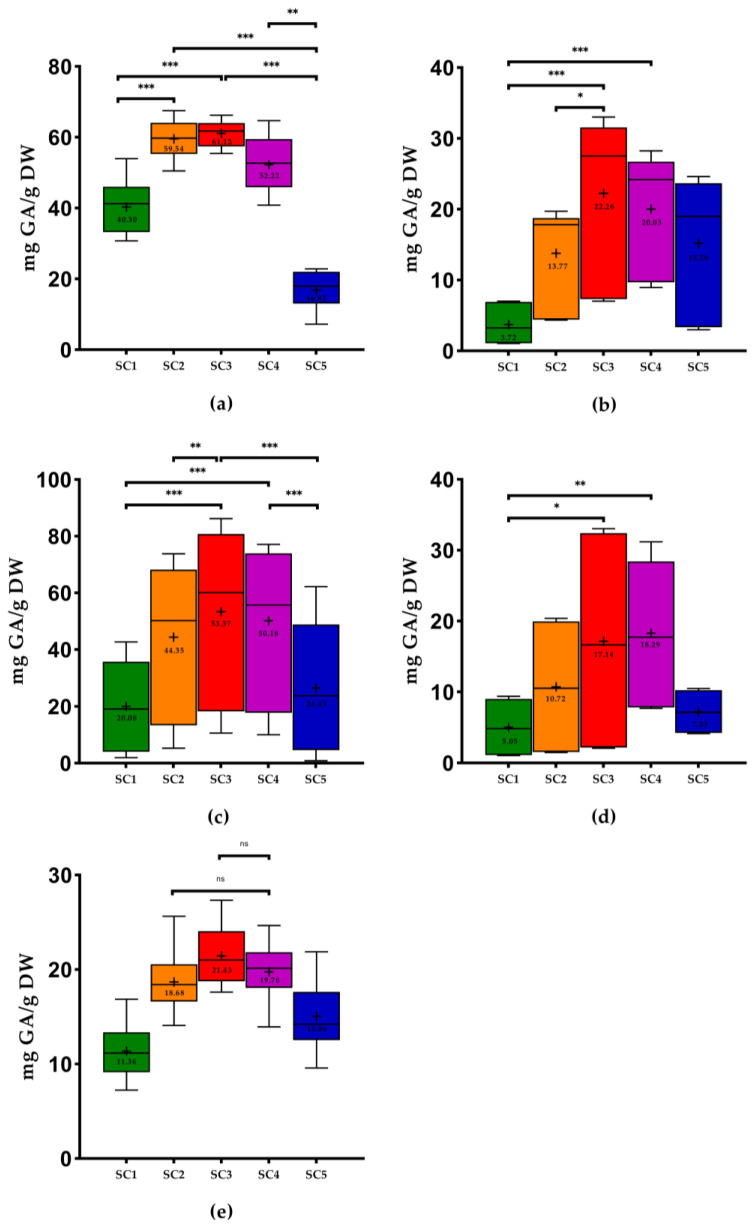
Box plots with quartiles (upper values 75%, median 50%, and lower values 25%) and mean values of TPC. GA, gallic acid; DW, dry weight; SC1, 100% H_2_O; SC2, EtOH: H_2_O (25:75, *v*/*v*); SC3, EtOH: H_2_O (50:50, *v*/*v*); SC4, EtOH: H_2_O (75:25, *v*/*v*); SC5, 100% EtOH; +, mean; (**a**) stems; (**b**) pomace; (**c**) seeds; (**d**) wine lees; and (**e**) grapevine shoots. * Significant at *p* < 0.05; ** significant at *p* < 0.01; *** significant at *p* < 0.001, ns—no significant, according to Friedman test.

**Figure 2 molecules-28-06660-f002:**
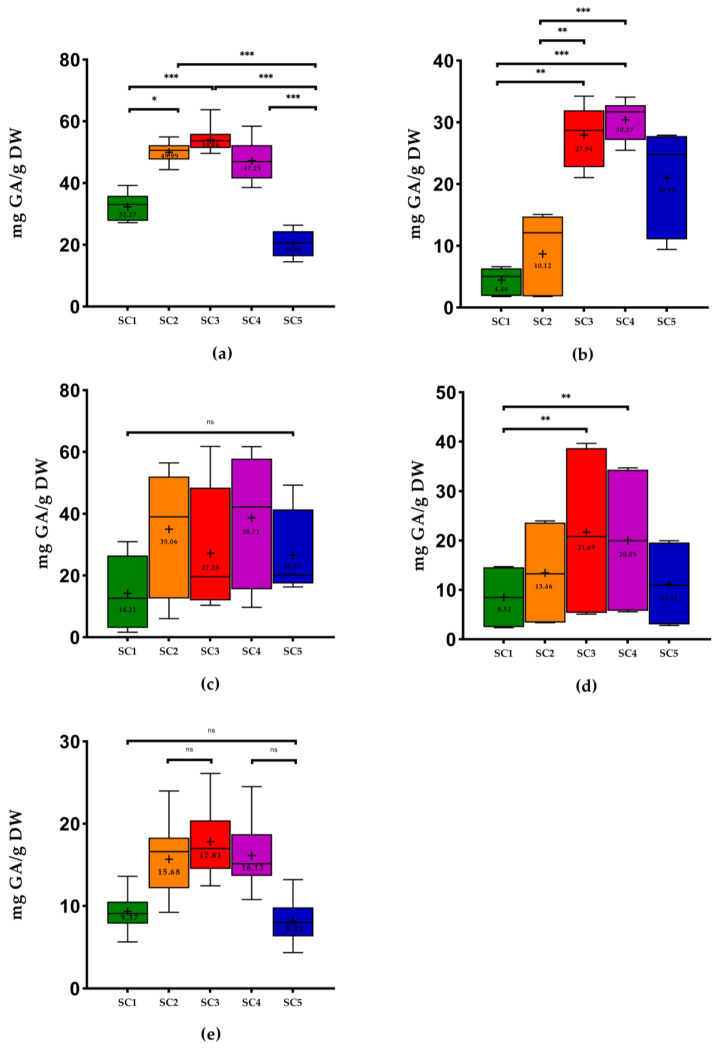
Box plots with quartiles (upper values 75%, median 50%, and lower values 25%) and mean values of ODC. GA, gallic acid; DW, dry weight; SC1, 100% H_2_O; SC2, EtOH: H_2_O (25:75, *v*/*v*); SC3, EtOH: H_2_O (50:50, *v*/*v*); SC4, EtOH: H_2_O (75:25, *v*/*v*); SC5, 100% EtOH; +, mean; (**a**) stems; (**b**) pomace; (**c**) seeds; (**d**) wine lees; and (**e**) grapevine shoots. * Significant at *p* < 0.05; ** significant at *p* < 0.01; *** significant at *p* < 0.001, ns—no significant, according to Friedman test.

**Figure 3 molecules-28-06660-f003:**
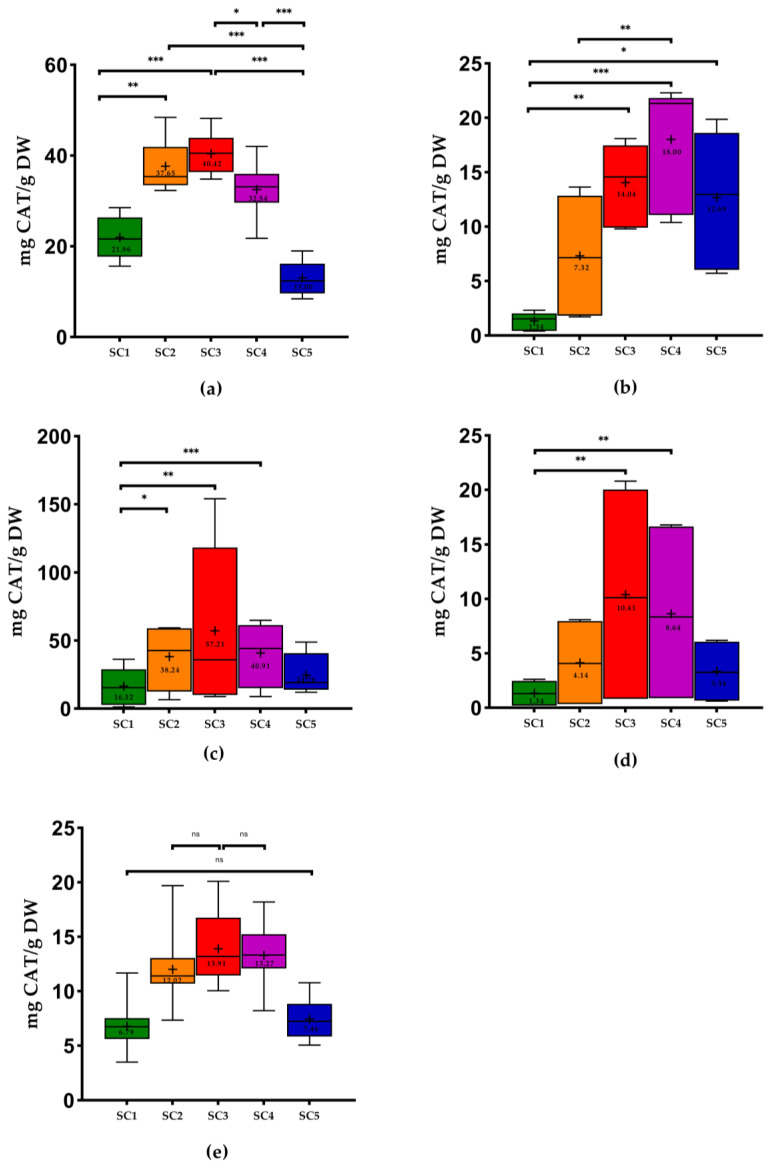
Box plots with quartiles (upper values 75%, median 50%, and lower values 25%) and mean values of FC. CAT, catechin; DW, dry weight; SC1, 100% H_2_O; SC2, EtOH: H_2_O (25:75, *v*/*v*); SC3, EtOH: H_2_O (50:50, *v*/*v*); SC4, EtOH: H_2_O (75:25, *v*/*v*); SC5, 100% EtOH; +, mean; (**a**) stems; (**b**) pomace; (**c**) seeds; (**d**) wine lees; and (**e**) grapevine shoots. * Significant at *p* < 0.05; ** significant at *p* < 0.01; *** significant at *p* < 0.001, ns—no significant, according to Friedman test.

**Figure 4 molecules-28-06660-f004:**
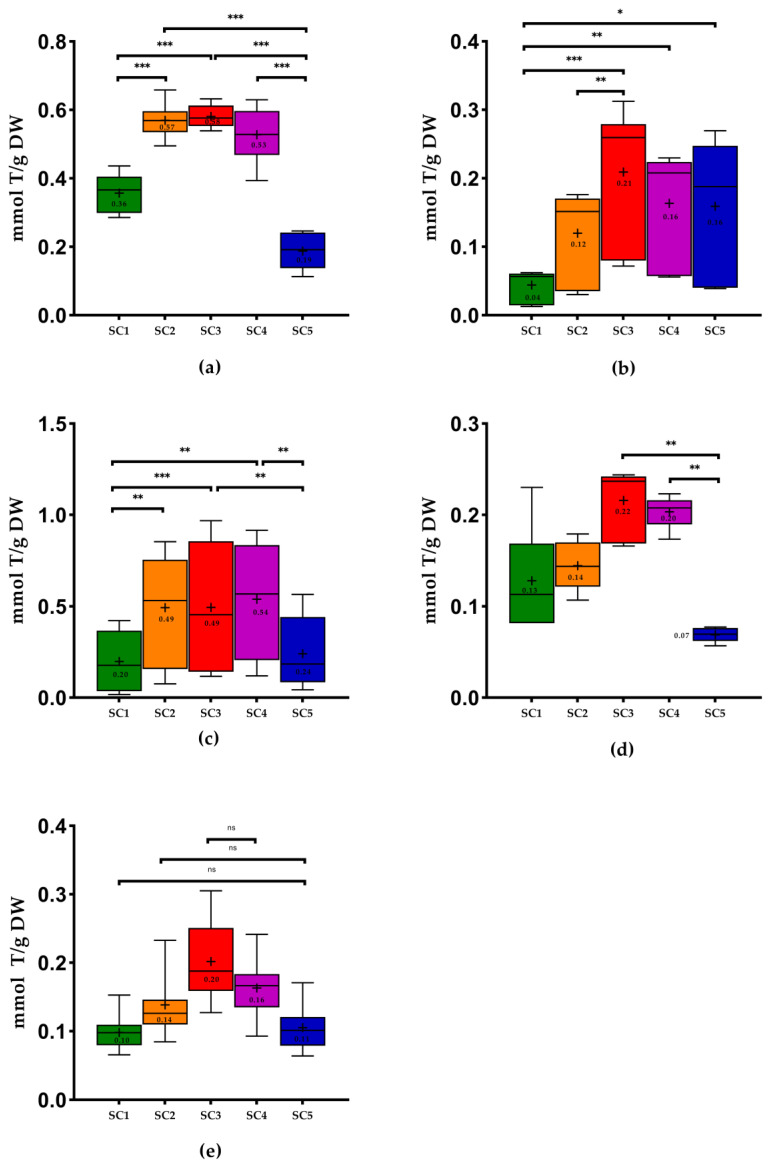
Box plots with quartiles (upper values 75%, median 50%, and lower values 25%) and mean values of FRAP assay. T, Trolox; DW, dry weight; SC1, 100% H_2_O; SC2, EtOH: H_2_O (25:75, *v*/*v*); SC3, EtOH: H_2_O (50:50, *v*/*v*); SC4, EtOH: H_2_O (75:25, *v*/*v*); SC5, 100% EtOH; +, mean; (**a**) stems; (**b**) pomace; (**c**) seeds; (**d**) wine lees; and (**e**) grapevine shoots. * Significant at *p* < 0.05; ** significant at *p* < 0.01; *** significant at *p* < 0.001, ns—no significant, according to Friedman test.

**Figure 5 molecules-28-06660-f005:**
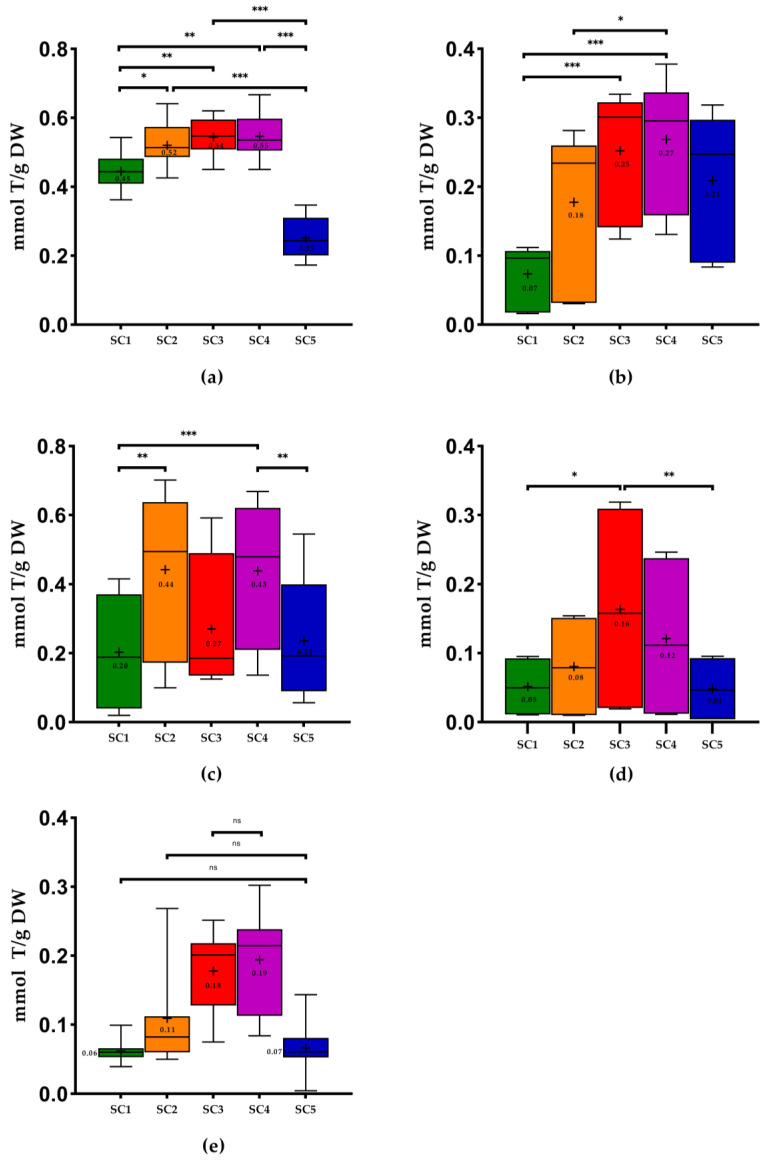
Box plots with quartiles (upper values 75%, median 50%, and lower values 25%) and mean values of DPPH assay. T, Trolox; DW, dry weight; SC1, 100% H_2_O; SC2, EtOH: H_2_O (25:75, *v*/*v*); SC3, EtOH: H_2_O (50:50, *v*/*v*); SC4, EtOH: H_2_O (75:25, *v*/*v*); SC5, 100% EtOH; +, mean; (**a**) stems; (**b**) pomace; (**c**) seeds; (**d**) wine lees; and (**e**) grapevine shoots. * Significant at *p* < 0.05; ** significant at *p* < 0.01; *** significant at *p* < 0.001, ns—no significant, according to Friedman test.

**Figure 6 molecules-28-06660-f006:**
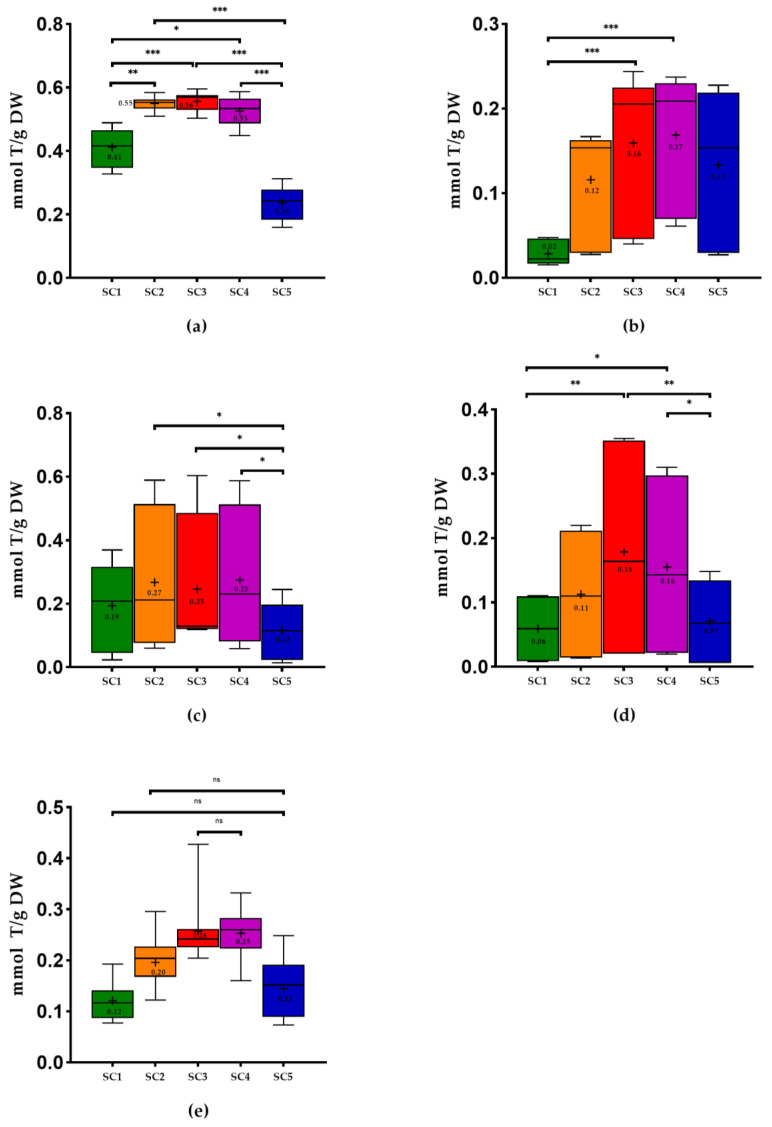
Box plots with quartiles (upper values 75%, median 50%, and lower values 25%) and mean values of ABTS assay. T, Trolox; DW, dry weight; SC1, 100% H_2_O; SC2, EtOH: H_2_O (25:75, *v*/*v*); SC3, EtOH: H_2_O (50:50, *v*/*v*); SC4, EtOH: H_2_O (75:25, *v*/*v*); SC5, 100% EtOH; +, mean; (**a**) stems; (**b**) pomace; (**c**) seeds; (**d**) wine lees; and (**e**) grapevine shoots. * Significant at *p* < 0.05; ** significant at *p* < 0.01; *** significant at *p* < 0.001, ns—no significant, according to Friedman test.

**Figure 7 molecules-28-06660-f007:**
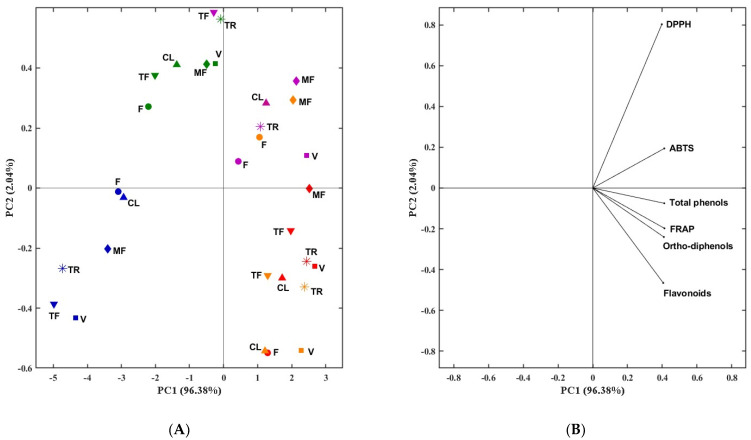
Principal component analysis (PCA) scores (**A**) and loadings plot (**B**) of phenolic content (TPC, ODC, and FC) and antioxidant capacity (FRAP, DPPH, and ABTS) of stems. Each color corresponds to solvents combinations (green: SC1: 100% H_2_O; orange: SC2: EtOH: H_2_O (25:75, *v*/*v*); red: SC3: EtOH: H_2_O (50:50, *v*/*v*); pink: SC4: EtOH: H_2_O (75:25, ***v*/*v***); blue: SC5: 100% EtOH. TPC, total phenolic content; ODC, *ortho*-diphenols content; FC, flavonoids content; FRAP, ferric-reducing antioxidant power; DPPH, scavenging capacity of DPPH radical; ABTS, scavenging capacity of ABTS radical.

**Figure 8 molecules-28-06660-f008:**
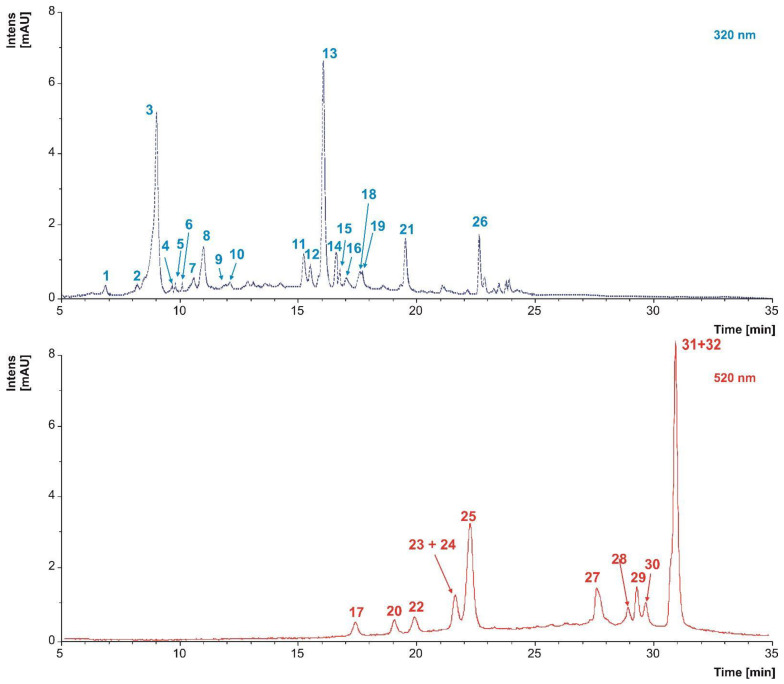
Representative HPLC–DAD chromatogram of Tinta Roriz variety, corresponding to two distinct wavelengths: 320 nm (nanometers) (depicted in blue), which highlights phenolic acids, stilbenes, flavonoids, and proanthocyanidins, and 520 nm (depicted in red), which represents anthocyanins.

**Figure 9 molecules-28-06660-f009:**
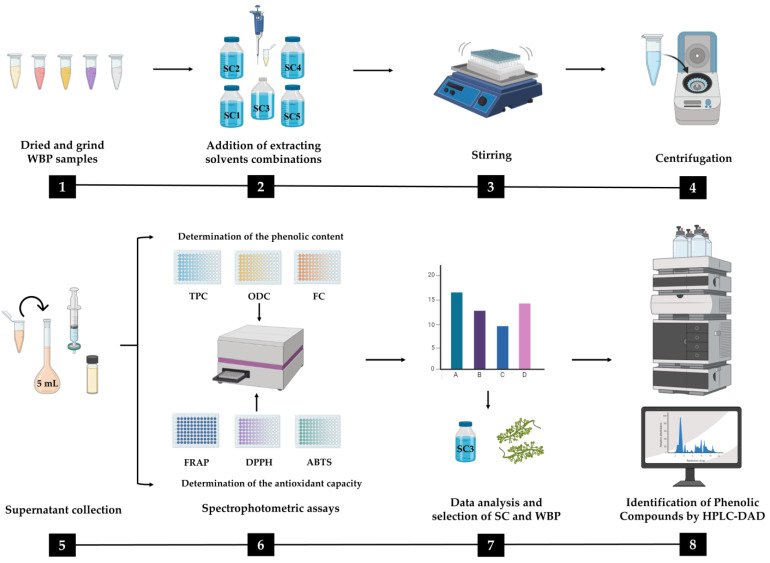
Schematic representation of the methodology used in this study. WBP, winery by-products; TPC, total phenolic content; ODC, *ortho*-diphenols content; FC, flavonoids content; FRAP, ferric-reducing antioxidant power; DPPH, 2,2-diphenyl-1-picrylhydrazyl; ABTS, 2,2-azino-bis (3-ethylbenzothiazoline-6-sulphonic acid) diammonium salt; HPLC, high-performance liquid chromatography with diode array detection; SC, solvent combination.

**Table 1 molecules-28-06660-t001:** Spectrophotometric assays of stem varieties hydroethanolic extracts (50:50, *v*/*v*).

Spectrophotometric Assays	White-Stem Varieties				Red-Stem Varieties	
	F	CL	V	MF	TF	TR
TPC (mg GA/g DW)	59.51 ± 3.88 ^ab^	60.35 ± 1.63 ^ab^	64.43 ± 1.34 ^a^	61.54 ± 1.46 ^ab^	56.17 ± 1.16 ^b^	64.73 ± 1.40 ^a^
ODC (mg GA/g DW)	50.65 ± 0.93 ^b^	53.30 ± 2.29 ^ab^	55.73 ± 0.68 ^ab^	58.93 ± 4.17 ^a^	54.08 ± 0.81 ^ab^	51.69 ± 0.80 ^b^
FC (mg CAT/g DW)	35.62 ± 1.15 ^b^	36.89 ± 1.93 ^b^	45.09 ± 1.09 ^a^	40.66 ± 3.01 ^ab^	40.52 ± 0.05 ^ab^	43.76 ± 3.92 ^a^
FRAP (mmol T/g DW)	0.55 ± 0.01 ^c^	0.56 ± 0.02 ^bc^	0.62 ± 0.01 ^a^	0.59 ± 0.02 ^abc^	0.55 ± 0.01 ^c^	0.61 ± 0.01 ^ab^
DPPH (mmol T/g DW)	0.46 ± 0.01 ^c^	0.51 ± 0.01 ^bc^	0.58 ± 0.04 ^ab^	0.60 ± 0.03 ^a^	0.55 ± 0.04 ^ab^	0.56 ± 0.03 ^ab^
ABTS (mmol T/g DW)	0.51 ± 0.01 ^b^	0.53 ± 0.01 ^b^	0.58 ± 0.00 ^a^	0.57 ± 0.00 ^a^	0.57 ± 0.01 ^a^	0.58 ± 0.01 ^a^

TPC, total phenolic content; ODC, *ortho*-diphenols content; FC, flavonoids content; FRAP, ferric-reducing antioxidant power; DPPH, scavenging capacity of DPPH radical; ABTS, scavenging capacity of ABTS radical; DW, dry weight. Stems varieties: F, Folgasão; CL, Códega do Larinho; V, Verdelho; MF, Malvasia Fina; TF, Touriga Franca; TR, Tinta Roriz. The values are represented as mean ± standard deviation (n = 3). An analysis of variance (ANOVA) was performed followed by a post hoc Tukey. Different letters correspond to significant differences between each variety (*p* < 0.05).

**Table 2 molecules-28-06660-t002:** Phenolic compounds identified in stems extracts.

Class of Compounds	Peak Number *	Rt	Identified Compounds	*Vitis vinifera* L. Stems Varieties					
				F	CL	V	MF	TF	TR
Phenolic Acids	**1**	6.9	Protocatechuic acid hexoside	X	X	<LOD ^Y^	X	X	X
	**3**	8.9	*trans*-caftaric acid	X	X	X	X	X	X
Stilbenes	**16**	17.1	Oxyresveratrol	X	X	X	X	<LOD	X
	**19**	17.8	*trans*-piceid	<LOD	<LOD	<LOD	X	X	X
	**26**	22.8	ԑ-viniferin	X	X	X	X	<LOD	X
Flavanols	**6**	10.2	Catechin	X	X	X	X	X	X
	**12**	15.4	Epicatechin gallate	X	X	X	X	<LOD	X
Flavonols	**9**	11.8	Quercetin-glucoside	<LOD	<LOD	<LOD	X	<LOD	X
	**11**	15.3	Quercetin-3-rutinoside	X	X	X	X	X	X
	**13**	16.0	Quercetin-3-*O*-glucuronide	X	X	X	X	X	X
	**14**	16.4	Kaempferol-3-*O*-glucoside	X	X	X	X	<LOD	<LOD
	**18**	17.6	Kaempferol-7-*O*-β-d-glucopyranoside	<LOD	<LOD	<LOD	X	X	X
Flavones	**15**	16.6	Luteolin-rutinoside	X	<LOD	X	X	<LOD	<LOD
Anthocyanins	**17**	17.4	Delphinidin-3-*O*-glucoside		<LOD	<LOD		<LOD	X
	**20**	19.0	Cyanidin-3-*O*-glucoside	<LOD	<LOD	<LOD	<LOD	<LOD	X
	**22**	20.0	Petunidin-3-*O*-glucoside	<LOD	<LOD	<LOD	<LOD	<LOD	<LOD
	**23**	21.7	Peonidin-3-*O*-glucoside ^W^	<LOD	<LOD	<LOD	<LOD	X	X
	**24**	21.7	Malvidin-3-*O*-glucoside ^W^	<LOD	<LOD	<LOD	<LOD	X	X
	**25**	22.1	Delphinidin-3-*O*-acetylglucoside	<LOD	<LOD	<LOD	<LOD	X	X
	**27**	27.8	Peonidin-3-*O*-acetylglucoside	<LOD	<LOD	<LOD	<LOD	X	X
	**28**	27.8	Malvidin-3-*O*-acetylglucoside	<LOD	<LOD	<LOD	<LOD	X	X
	**29**	29.4	Delphinidin-3-*O-p*-coumaroylglucoside	<LOD	<LOD	<LOD	<LOD	X	X
	**30**	29.9	Cyanidin-3-*O-p*-coumaroylglucoside	<LOD	<LOD	<LOD	<LOD	X	X
	**31**	31.0	Petunidin-3-*O-p*-coumaroylglucoside ^Z^	<LOD	<LOD	<LOD	<LOD	X	X
	**32**	31.0	Malvidin-3-*O*-*p*-coumaroylglucoside ^Z^	<LOD	<LOD	<LOD	<LOD	X	X
Proanthocyanidins	**2**	8.2	Proanthocyanidin dimer (B-type) Isomer 1	X	X	X	X	X	X
	**4**	9.5	Proanthocyanidin dimer (B-type) Isomer 2	<LOD	<LOD	X	X	<LOD	X
	**5**	9.8	Proanthocyanidin trimer (B-type) Isomer 1	X	X	X	X	<LOD	<LOD
	**7**	10.6	Proanthocyanidin dimer-gallate Isomer 1	X	X	X	X	X	X
	**8**	11.2	Proanthocyanidin dimer-gallate Isomer 2	X	X	X	X	X	X
	**10**	12.1	Proanthocyanidin trimer (B-type) Isomer 2	X	<LOD	X	X	X	<LOD
	**21**	19.5	Proanthocyanidin trimmer (B-type) monogallate	X	X	X	X	X	X

^Y^, LOD, limit of detection; ^W, Z^, coeluting compounds; *, peak number according to Tinta Roriz variety (Figure 8); Rt, retention time (minutes). Stems varieties: F, Folgasão; CL, Códega do Larinho; V, Verdelho; MF, Malvasia Fina; TF, Touriga Franca; TR, Tinta Roriz.

**Table 3 molecules-28-06660-t003:** Sampling of winery by-products from the *Região Demarcada do Douro*.

WBP	Samples Number	Type of Variety	Varieties	Harvest	Sampling Step	Sub-Regions
Stems	3	Single white variety	F, V, MF	2021	After destemming	CC
	1	Single white variety	CL	2021		BC
	2	Single red varieties	TF, TR	2021		CC
Pomace (Seeds, pulp, skins)	2	Single white variety	M	2021	Before fermentation	BC
		Single white variety	F	2021		CC
	1	Mixture of several red varieties	TN, TF, So	2021	After fermentation	DS
Seeds	2	Single white variety	M	2022	Before fermentation	BC
		Mixture of several white varieties	Vio, MF, FP	2022		BC, CC, DS
	2	Mixture of several red varieties	TN, TR, TB	2022	After fermentation	BC, CC, DS
		Mixture of several red varieties	TN, TB, TA, TR	2022		CC
Wine lees	1	Mixture of several white varieties	R, MF, Vio	2022	Post alcoholic fermentation	BC, CC, DS
	1	Mixture of several red varieties	TN, TF, TR	2022		BC, CC, DS
Grapevine shoots	10	Single white varieties	S, F, R, EC, A, Vio, CL, MR, MF, FP	2021	After pruning (lignified)	BC
	3	Single red varieties	TA, TR, TB	2021		BC

WBP, winery by-products; BC, Baixo Corgo sub-region; CC, Cima Corgo sub-region; DS, Douro Superior sub-region. Varieties: A, Arinto; CL, Códega do Larinho; EC, Esgana Cão; F, Folgasão; FP, Fernão Pires; M, Moscatel; MF, Malvasia Fina; MR, Malvasia Rei; R, Rabigato; S, Síria; So, Sousão; TA, Tinta Amarela; TB, Tinta Barroca; TF, Touriga Franca; TN, Touriga Nacional; TR, Tinta Roriz; V, Verdelho; Vio, Viosinho.

## Data Availability

Not applicable.

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
