# Peer review of "Exploring the Antioxidant Potential of Phenolic Compounds from Winery By-Products by Hydroethanolic Extraction"

_molecules, 2023, doi:10.3390/molecules28186660_

Round 1
Reviewer 1 Report
Dear authors, the research presents useful information as a first step to obtain bioactive compounds from winery by-products. Therefore, the manuscript can be accepted for publication in its present form.
Author Response
Dear Editor of Molecules,
In reply to the review performed on the paper entitled “Exploring the Antioxidant Potential of Phenolic Compounds from Winery By-Products by Hydroethanolic Extraction”, we would like to acknowledge the valuable comments performed by the editor that kindly accepted to revise our manuscript. We would like to confirm that we have addressed most issues and answered the questions made by the reviewers. We hope the answers below and modifications that have been done in the manuscript are clear and concise enough as required by the reviewer to enable the publication of the manuscript in Molecules.
Answer to referee’s comments and queries
Detailed responses to Reviewer 1
Reviewer´s comment: Dear authors, the research presents useful information as a first step to obtain bioactive compounds from winery by-products. Therefore, the manuscript can be accepted for publication in its present form.
Our reply: We express our appreciation for your thoughtful evaluation. Your recognition of the research as a valuable initial stride towards extracting bioactive compounds from winery by-products is truly encouraging. Based on your assessment, we are pleased to inform you that the manuscript has been improved.
Sincerely,
Ana Isabel Ramos Novo Amorim de Barros

Reviewer 2 Report
Here are some comments and suggestions for the given text:
- Clarify the purpose and context: It would be helpful to provide a brief introduction or background information about the study and its objectives. This will give readers a better understanding of the context and significance of the results.
- Improve readability: The text contains many abbreviations and acronyms (e.g., FRAP, DPPH, ABTS) without clear explanations. Consider providing a key or legend to explain these abbreviations, especially for readers who may not be familiar with the specific terminology.
- Use clear and concise language: Some sentences are long and complex, making it difficult to follow the main points. Try breaking them down into shorter sentences and use simpler language to enhance readability.
- Provide references: When citing previous studies (e.g., Jimenez-Moreno et al. [29]), it's important to include proper references to enable readers to access the original sources for more information.
- Clarify units and measurements: In several instances, the text mentions different units or measurements (e.g., mmol T/g DW). It would be helpful to clarify these units and provide a brief explanation to ensure readers understand the data being presented.
- Include figures and captions: The text refers to figures (e.g., Figure 6b) without actually including them. Consider adding the figures to provide visual representation of the data being discussed. Also, make sure to include clear captions for each figure to explain the content and context.
- Proofread for errors: There are some minor typographical errors and inconsistencies throughout the text. It's important to proofread the entire document to ensure accuracy and consistency in spelling, punctuation, and grammar.
No issues
Author Response
Dear Editor of Molecules,
In reply to the review performed on the paper entitled “Exploring the Antioxidant Potential of Phenolic Compounds from Winery By-Products by Hydroethanolic Extraction”, we would like to acknowledge the valuable comments performed by the editor that kindly accepted to revise our manuscript. We would like to confirm that we have addressed most issues and answered the questions made by the reviewers. We hope the answers below and modifications that have been done in the manuscript are clear and concise enough as required by the reviewer to enable the publication of the manuscript in Molecules.
Answer to referee’s comments and queries
Detailed responses to Reviewer 2
Reviewer´s comment: Clarify the purpose and context: It would be helpful to provide a brief introduction or background information about the study and its objectives. This will give readers a better understanding of the context and significance of the results.
Our reply: The authors are grateful for the reviewer's comments and insightful considerations. We agree with the reviewer´s decision on the points that need clarification and improvement. It was already corrected.
Reviewer´s comment: Improve readability: The text contains many abbreviations and acronyms (e.g., FRAP, DPPH, ABTS) without clear explanations. Consider providing a key or legend to explain these abbreviations, especially for readers who may not be familiar with the specific terminology.
Our reply: Thank you for your suggestion, it was already corrected.
Reviewer´s comment: Use clear and concise language: Some sentences are long and complex, making it difficult to follow the main points. Try breaking them down into shorter sentences and use simpler language to enhance readability.
Our reply: The authors appreciate the relevance of the comment. The article's writing has been completely revised to make the sentences clearer and more concise.
Reviewer´s comment: Provide references: When citing previous studies (e.g., Jimenez-Moreno et al. [29]), it's important to include proper references to enable readers to access the original sources for more information.
Our reply: Thank you for your suggestion. We have rephrased the sentences to enhance clarity. Specifically, the authors (Jimenez-Moreno et al. [29]) achieved improved results when using the solvent H2O: EtOH (50:50) in conjunction with stems.
Reviewer´s comment: Clarify units and measurements: In several instances, the text mentions different units or measurements (e.g., mmol T/g DW). It would be helpful to clarify these units and provide a brief explanation to ensure readers understand the data being presented.
Our reply: Thank you for your suggestion. The units used in this study align with the applied methodology. For example, for total phenols, we employ gallic acid as the standard, and it's expressed as mg GA/g of dry weight. Regarding the three antioxidant capacity methodologies, we utilize mmol mg T/g of dry weight, and Trolox serves as the standard.
The explanation of the units is in the sections 3.4.1, 3.4.2., 3.4.3, 3.5.1, 3.5.2, and 3.5.3.
Reviewer´s comment: Include figures and captions: The text refers to figures (e.g., Figure 6b) without actually including them. Consider adding the figures to provide visual representation of the data being discussed. Also, make sure to include clear captions for each figure to explain the content and context.
Our reply: The authors appreciate the relevance of the comment, as this particular section was indeed excessively detailed. Following the suggestion, we have rewritten this section in a more concise manner.
Reviewer´s comment: Proofread for errors: There are some minor typographical errors and inconsistencies throughout the text. It's important to proofread the entire document to ensure accuracy and consistency in spelling, punctuation, and grammar.
Our reply: Thank you for your suggestion. We have addressed the errors and inconsistencies, and we've also rectified the table and figure numbering throughout the text.
Sincerely,
Ana Isabel Ramos Novo Amorim de Barros

Reviewer 3 Report
The work deals with characterizing different types of wine by-products, obtained from different types of varieties of grapes. The topic is interesting because reinforces the perspective of using by-products from wineries to obtain bioactive and antioxidant compounds. Nevertheless, the wording has several flaws, starting with a lack of discussion of results. This section includes a lot of experimental data related to the contents of phenolics and the antioxidant activities (ABTS, DPPH, FRAP), but the wording only states a description of trends based on the solvent with the highest yield of extraction and a list of paragraphs that seems to have published the same trends for the same types of samples but from different regions, without any explanation about the effect of extraction conditions, the type of variety, or the type of by-product used as a source of antioxidants on the extraction yielding. Conclusions are stated more as a perspective and future activities, rather than including the most relevant contribution to knowledge generated from this work.
Other specific comments are given below:
L30. It is recommended to consult Annual sources to establish these quantities, i.e. based on the report of the OIV Annual Assessment of the World Vine and Wine Sector 2021 (www.oiv.int), “the world's vineyards produced a total of 74.8 mt of fresh grapes in 2021, where 37.2 million tonnes were processed as pressed grapes (34.1 million tonnes for production of wine and 3.1 million tonnes for the production of musts and juices”. These values are much higher than that indicated in these lines.
L75-80. It is suggested to indicate the different types of varieties or regions considered in the study, as well as their nomenclature definition.
Figure captions. Because there are several nomenclatures and abbreviations in the axis referring to each variety, it is suggested that the description of the Figure use the same order as that in the plot.
Figures 1-3 contain a lot of information becoming difficult to follow and to identify trends, it is suggested to add legend captions in the plots for indicating the type of solvent used in each group, it would be easier to locate the effect of the solvent on the extraction of the different compounds, instead of reading a very large figure caption with a lot of abbreviations. This suggestion can be changed for another that simplifies the reading of data.
L109-130. Add the standard deviations for the mean values.
Sections 2.1 and 2.2. The discussion of results in terms of comparing data from this work to that already published elsewhere should be improved by directing it to what are the implications, perspectives, explanations about differences, etc., instead of giving a large list of works that reported higher or lower contents of bioactive compounds using similar o the same extraction conditions, otherwise, why testing different ethanol concentrations is there plenty of works that indicate that ethanol around 50% is the best extraction solvent.
Table 1. This table can be omitted since these values are already contained in Figure 7, instead, you can provide a more in-depth discussion about the effect of solvent and the recovery of TPC, TFC, and antioxidant activities.
Section 3.2 Include the geographical coordinates for each winery. Perhaps you could consider using a Table for defining each one of the samples studied in this work.
L458. Revise the symbols used in the units of temperature.
L464. Explain why was the solvent acidified just for grape pomace and wine lees. How could this acidification affect the yielding in the rest of the samples?
Table 2. It is suggested to add the abbreviations used for each variety.
Information regarding the preliminary identification of the main compounds by HPLC in each extract should be included.
Include information regarding the conditions used for processing the varieties of grapes, since they were obtained from two different wineries, and there is any information about the preliminary processing prior to the recovery by the authors.
The language in the current manuscript is adequate, some typos were found. Nevertheless, it is essential to include a more in-depth discussion, rather than presenting a mere description of data and comparing the information from the literature without any explanation derived from such comparison.
Author Response
Dear Editor of Molecules,
In reply to the review performed on the paper entitled “Exploring the Antioxidant Potential of Phenolic Compounds from Winery By-Products by Hydroethanolic Extraction”, we would like to acknowledge the valuable comments performed by the editor that kindly accepted to revise our manuscript. We would like to confirm that we have addressed most issues and answered the questions made by the reviewers. We hope the answers below and modifications that have been done in the manuscript are clear and concise enough as required by the reviewer to enable the publication of the manuscript in Molecules.
Answer to referee’s comments and queries
Detailed responses to Reviewer 3
Reviewer´s comment: L30. It is recommended to consult Annual sources to establish these quantities, i.e. based on the report of the OIV Annual Assessment of the World Vine and Wine Sector 2021 (www.oiv.int), “the world's vineyards produced a total of 74.8 mt of fresh grapes in 2021, where 37.2 million tonnes were processed as pressed grapes (34.1 million tonnes for production of wine and 3.1 million tonnes for the production of musts and juices”. These values are much higher than that indicated in these lines.
Our reply: The authors are grateful for the reviewer's comments. We fully agree with this pertinent comment, and we would like to inform you that this topic has been completely revised and corrected.
Reviewer´s comment: L75-80. It is suggested to indicate the different types of varieties or regions considered in the study, as well as their nomenclature definition.
Our reply: The authors appreciate the comment and would like to inform you that this topic has been revised to clearly clarified.
Reviewer´s comment: Figure captions. Because there are several nomenclatures and abbreviations in the axis referring to each variety, it is suggested that the description of the Figure use the same order as that in the plot.
Our reply: Thank you for your suggestion. The figures have been all changed, and improved.
Reviewer´s comment: Figures 1-3 contain a lot of information becoming difficult to follow and to identify trends, it is suggested to add legend captions in the plots for indicating the type of solvent used in each group, it would be easier to locate the effect of the solvent on the extraction of the different compounds, instead of reading a very large figure caption with a lot of abbreviations. This suggestion can be changed for another that simplifies the reading of data.
Our reply: Thank you for your suggestion. The figures have been all changed, and improved.
Reviewer´s comment: L109-130. Add the standard deviations for the mean values.
Our reply: Thank you for your suggestion. It was already changed.
Reviewer´s comment: Sections 2.1 and 2.2. The discussion of results in terms of comparing data from this work to that already published elsewhere should be improved by directing it to what are the implications, perspectives, explanations about differences, etc., instead of giving a large list of works that reported higher or lower contents of bioactive compounds using similar or the same extraction conditions, otherwise, why testing different ethanol concentrations is there plenty of works that indicate that ethanol around 50% is the best extraction solvent.
Our reply: Thank you for your suggestion. It was already changed.
Reviewer´s comment: Table 1. This table can be omitted since these values are already contained in Figure 7, instead, you can provide a more in-depth discussion about the effect of solvent and the recovery of TPC, TFC, and antioxidant activities.
Our reply: Thank you for your suggestion. The table has been removed.
Reviewer´s comment: Section 3.2 Include the geographical coordinates for each winery. Perhaps you could consider using a Table for defining each one of the samples studied in this work.
Our reply: Our reply: The authors are grateful for the reviewer's comments; however, the samples of winery by-products come from several vineyards in the Região Demarcada do Douro (Cima Corgo, Baixo Corgo, and Douro Superior). For example, the mixture of several red grape pomaces could be from different vineyards. However, each sub-region of the Região Demarcada do Douro for each sample was added to a new Table in Material and Methods Section.
Reviewer´s comment: L458. Revise the symbols used in the units of temperature.
Our reply: Thank you for your suggestion. It was already changed.
Reviewer´s comment: L464. Explain why was the solvent acidified just for grape pomace and wine lees. How could this acidification affect the yielding in the rest of the samples?
Our reply: Thank you for your question. The solvent was acidified specifically for grape pomace and wine lees due to the presence of anthocyanins in these samples that require this adjustment. Acidification helps in the extraction of target compounds from these materials. However, it's important to consider the potential impact on other samples. Acidification might affect the yield in other samples by altering the pH-dependent solubility of compounds. If the pH is not optimized for a particular sample, it could lead to incomplete extraction of the desired compounds, resulting in lower yields. It's essential to carefully control the acidification process to ensure that it doesn't negatively impact the extraction efficiency in non-acidified samples.
Reviewer´s comment: Table 2. It is suggested to add the abbreviations used for each variety.
Our reply: Thank you for your suggestion. It was already changed.
Reviewer´s comment: Information regarding the preliminary identification of the main compounds by HPLC in each extract should be included.
Our reply: Thank you for your suggestion. The identification of the phenolic compounds by HPLC-DAD was performed and added to the manuscript.
Reviewer´s comment: Include information regarding the conditions used for processing the varieties of grapes, since they were obtained from two different wineries, and there is any information about the preliminary processing prior to the recovery by the authors.
Our reply: Thank you for your suggestion. We don’t have access to this type of data. The samples were provided from different vineyards of sub-regions of the Região Demarcada do Douro.
Sincerely,
Ana Isabel Ramos Novo Amorim de Barros

Reviewer 4 Report
The document deals with a highly topical issue of great interest to the viticulture and oenology industry. The authors have done a great experimental work that should be published, as it can be of great help in future researches.
However, before proceeding to publication, in my opinion the authors need a major revision, both in the materials and methods section and in the expression of the results obtained.
MATERIALS AND METHODS
The authors mentioned that they followed the Abraao et al’, metodology but
Have the authors taken into account the differences in phenolic content and distribution between the starting material (Prunus lusitanica L) and the grapes? How did the authors ensure that all content or a significant amount of phenolic material was extracted from each of the samples?
Besides, in my opinion it is necessary to specify in the material an method section
· Extraction time,
· Number of repeated extractions with each of the ethanol-water mixtures used?
On other hand, please also specify when were each of the samples taken the following:
· The stems, immediately after the processing of the bunches?
· How were the seeds extracted from the samples?
· In the case of the shoots: in what state were they in, green or already lignified?
· Also specify the state of the pommace. In the case of the white varieties, was it collected before vinification? and the red varieties: at what time?
I propose some changes in Table 2
|
|
Table 2. Sampling of winery by-products from the Douro region |
|
|
Winery by- Total |
Varieties Harvest Cellar/Location/ |
|
RESULTS
· The Figures 1, 2, 3, 4, 5 and 6 are very difficult to read and understand. In fact, in black and white it is impossible to see the results!. The authors should employ and display their results on other types of graphs (perhaps bar graphs?) or even tables.
· In the same way, in the Figure 7 (PCA) the authors have to made bigger the simbols . Perhaps, they should change the symbols by codes ( ie: FSC1: folgasao, 100% H20 etc…)
Thus, I could not properly read and review the results section.

Author Response
Dear Editor of Molecules,
In reply to the review performed on the paper entitled “Exploring the Antioxidant Potential of Phenolic Compounds from Winery By-Products by Hydroethanolic Extraction”, we would like to acknowledge the valuable comments performed by the editor that kindly accepted to revise our manuscript. We would like to confirm that we have addressed most issues and answered the questions made by the reviewers. We hope the answers below and modifications that have been done in the manuscript are clear and concise enough as required by the reviewer to enable the publication of the manuscript in Molecules.
Answer to referee’s comments and queries
Detailed responses to Reviewer 4
Reviewer´s comment: The authors mentioned that they followed the Abraão et al., methodology but have the authors taken into account the differences in phenolic content and distribution between the starting material (Prunus lusitanica L) and the grapes? How did the authors ensure that all content or a significant amount of phenolic material was extracted from each of the samples?
Our reply: Thank you for your question. The extraction methodology for phenolic compounds is consistent within our research group, with minor variations based on the specific sample being used, primarily in terms of the water-to-solvent ratio. Abraão et al. are also part of our research group and employed a similar solvent to our study, albeit with a different water-to-ethanol ratio (70:30, v/v). However, since they worked with a distinct matrix, Abraão et al. found that this proportion provided the optimal yield, in contrast to our study, which generally achieved better results using a water-to-ethanol ratio of 50:50, v/v.
Reviewer´s comment: Besides, in my opinion it is necessary to specify in the material and method section:
- Extraction time,
- Number of repeated extractions with each of the ethanol-water mixtures used?
Our reply: Thank you for your helpful input. We've made the necessary changes based on your suggestion. For each individual extraction, the duration was 45 minutes, with 30 minutes in the agitator and an additional 15 minutes in the centrifuge. We conducted a total of 3 repeated extractions. When considering all 3 extractions, the cumulative time is 2 hours and 15 minutes. This additional information has been incorporated into Section 3.3, specifically in lines 481-492.
Reviewer´s comments: On other hand, please also specify when were each of the samples taken the following:
The stems, immediately after the processing of the bunches?
Our reply: Thank you for your question. The stems were collected after destemming. This information was already added.
Reviewer´s comment: How were the seeds extracted from the samples?
Our reply: Thank you for your question. The seeds were extracted according to the protocol of our working group referred to in section 3.3.
Reviewer´s comment: In the case of the shoots: in what state were they in, green or already lignified?
Our reply: Thank you for your question. This information was already added. The grapevine-shoots were collected lignified.
Reviewer´s comment: Also specify the state of the pomace. In the case of the white varieties, was it collected before vinification? and the red varieties: at what time?
Our reply: Thank you for your question. This information was already added in the manuscript. In the case of white varieties, the pomace was collected before vinification, and in the red grape varieties after the vinification process.
Reviewer´s comments: The Figures 1, 2, 3, 4, 5 and 6 are very difficult to read and understand. In fact, in black and white it is impossible to see the results! The authors should employ and display their results on other types of graphs (perhaps bar graphs?) or even tables.
Our reply: Thank you for your suggestion. The results were already changed in new Figures, to improve the interpretation of the results.
Reviewer´s comments: In the same way, in the Figure 7 (PCA) the authors have to made bigger the symbols. Perhaps, they should change the symbols by codes (ie: FSC1: Folgasão, 100% H2O etc…)
Our reply: Thank you for your suggestion. The symbols by codes were improved.
Sincerely,
Ana Isabel Ramos Novo Amorim de Barros

Round 2
Reviewer 2 Report
the manuscript has already been improved. in my opinion, this manuscript can be accepted
no issues
Reviewer 4 Report
I would like to congratulate the authors. They substantially improved the manuscript, both in form and content.
They should only review carefully some remarks concerning the numeration of figures and tables. I found some errors that I indicate in the attached document.
